# Race: How the Post-Genomic Era Has Unmasked a Misconception Promoted by Healthcare

**DOI:** 10.3390/medicina59050861

**Published:** 2023-04-28

**Authors:** Donna Schaare, Ludovico Abenavoli, Luigi Boccuto

**Affiliations:** 1Ph.D. Program in Healthcare Genetics and Genomics, School of Nursing, College of Behavioral, Social and Health Sciences, Clemson University, Clemson, SC 29634, USA; dschaar@g.clemson.edu; 2Department of Health Sciences, University “Magna Graecia”, 88100 Catanzaro, Italy; l.abenavoli@unicz.it

**Keywords:** human genome project, race, healthcare, policy, post-genomic era

## Abstract

The term “race” has been employed to categorize human beings into distinct groups based on some perceived biological distinctions. This concept was debunked with the completion of the Human Genome Project and its revolutionary findings that all humans are >99% genetically identical, subsequently making the term “race” obsolete. Unfortunately, the previous misconception is being propagated by the continued use of the term to capture demographic information in healthcare in an attempt to improve equity. This paper seeks to review the history of the term “race”, analyze the current policy, and discuss its limitations. It is important to note that our analysis was exclusively focused on the United States healthcare system and the Affordable Care Act; as such, it may not reflect other regions’ policies, including those in Africa, Asia, and the Middle East. However, we feel that this policy analysis may serve as a model to recommend alterations that mirror the post-genomic era. The need for this policy change was recently highlighted in the 2022 ASHG presidential address, One Human Race: Billions of Genomes, and will reflect the knowledge gleaned by the scientific community through the conclusions of the Human Genome Project.

## 1. Introduction

“Race” has been defined by Ford and Kelly as “a social construct based on phenotypic genetic expression” [1]. Although the article emphasizes that it is a social construct, the unfortunate fact is that most of society believes that “race” is biologically real [2,3,4,5]. Politics employs this construct to determine government policy and civil rights laws. Additionally, with the recent passing of the Affordable Care Act (ACA), healthcare has become a facilitator too, by requiring “race” to be declared on all health intake forms per Section 4302(a) [6]. These policies continue to mask the breakthrough findings of the Human Genome Project that “race” is not real and, instead, reinforce the misconception that “race” has a genetic component [2,3,4,5]. The collection of “racial” data in healthcare supports the erroneous concept of “race” as a biological concept. This is an inaccuracy that healthcare is fostering throughout society.

For years, health disparities have been linked to physiologic genetic factors and, more recently, socio-economic and socio-political causes. Inequities have been apparent in the United States for over a hundred years, as first summarized in the 1985 Heckler Report. The report resulted in the formation of the Office of Minority Health (OMH), a branch of the US Department of Health and Human Services (HHS). HHS is a government agency that is accountable for safeguarding the health of US residents and offering them services, especially to the underserved. The OMH is responsible for creating health policies and programs that improve the health of minority populations to reduce health disparities. The establishment of the OMH was followed by 25 years of investigation to comprehend gaps in medical access, utilization, and health outcomes stratified by “race”. The study’s conclusion uncovered a critical limitation: a lack of standardized data on self-reported “race” and ethnicity [7].

Although not captured in the healthcare setting, “race” has been collected in the United States since 1790 through decennial censuses [8]. The process was formalized in 1977, when the Office of Management and Budget (OMB), under a decree from the Federal Government, issued Statistical Policy Directive No. 15. In the United States, the OMB is responsible for monitoring the effectiveness of government programs. This OMB policy established the Race and Ethnic Standards for Federal Statistics and Administrative Reporting for improved civil rights monitoring, prosecution, and other governmental program recordings [9]. The socio-political–legal environment at the time of the 1977 OMB standards was motivated by the climate of the post-Vietnam War and the evolving civil rights movement. It would take until the late 1990s and early 2000s to begin to investigate the more significant obstacles to health and the disproportionate outcomes seen in minority populations. In 1997, after two years of meetings and analysis, the OMB revised the standards to reflect the nation’s growing diversity. By the early 2000s, multiple studies had documented profound health disparities across the country. There were increased cries for more accurate data, combined with standardizing the collection of data, and a better understanding of the obstacles related to “race” [6]. The culmination came in 2009, with the formation of the Subcommittee on Standardized Collection of Race/Ethnicity Data for Healthcare Quality, under the OMB, as requested by the Agency for Healthcare Research and Quality. The agency serves as the research arm of the HHS, designed to improve the efficiency of healthcare services. Their report highlighted “that the lack of standardized data relevant to race, ethnicity, and language diminished the likelihood that effective actions could be identified to reduce specific health disparities” [6] (p. 125). 

This report was released just one year before the passage of the ACA. Therefore, policymakers were aware of the mandate to combat health disparities and the need for reliable “race” data was required. Section 4302(a) of the ACA, passed on 23 March 2010, requires that “race” be captured by Healthcare Providers, as established by the Secretary of the US Department of Health and Human Services (HHS) data collection standards in October 2011 [7]. The requirement of “race” collection was facilitated through paper, self-reported patient questionnaires, which are similar to the attached question (Table 1).

The development of the questionnaire, as stated earlier, was the responsibility of the Secretary of HHS. It was initially based on the OMB 1997 standards but added additional options of Asian and Native Hawaiian/Pacific Islander categories for more granularity [6,10,11]. Never before had “race” and ethnicity collection been required in healthcare [6]. This policy was implemented in October 2011 and it is expected of every healthcare provider who accepts federally funded programs, including Medicare and Medicaid and any commercial plan purchased under the ACA. All patients are thus screened, and their self-reported “race” is entered into their Electronic Health Record (EHR), which has been mandatory since 1 January 2014. Therefore, based on the new policy, “race” should be captured for every patient throughout the United States by employing these tools or resources, not only in clinicians’ offices but also in hospitals and any healthcare setting. 

## 2. Policy Analysis

Policy evaluation is an iterative process that requires policymakers to periodically review the policy to determine what is working and what needs improvement. The analysis conducted in this commentary paper considered solely the United States healthcare system and the Affordable Care Act; as such, it may not reflect other regions’ policies, including those in Africa, Asia, and the Middle East. 

Due to the inequities highlighted during the global pandemic, the newly appointed Biden administration executed a Section 4302(a) policy evaluation. The policy’s goal to capture “racial” data to track healthcare inequalities is its benefit. However, the resulting limitations speak to the failure of its execution. “The COVID-19 pandemic exposed our long-standing inability to collect, share, and act on meaningful race and ethnicity data in health care. According to the Centers for Disease Control and Prevention, race and ethnicity data are not available for nearly 40 percent of people testing positive for COVID or receiving a vaccine… These gaps limit our knowledge of health disparities and our ability to eliminate them.” [11]. So clearly, the policy has not been able to fix the problem. It has not even succeeded in achieving the first goal of collecting accurate data on “race”, which makes the second goal of tracking the data to shape future health policies to reduce disparities impossible to consider. Due to these findings, the Biden administration requested the constitution of an independent commission consisting of Grantmakers in Health and the National Committee for Quality Assurance. The commission reviewed the policy’s shortcomings and uncovered multiple barriers to its success, including inconsistencies in “racial” data within various health programs due to a lack of consistency in the use of standards (Table 2) [7]. 

Fear of discrimination or persecution from a history of medical abuse is another obstruction in the completion of “racial” data [11]. “Race” and healthcare have had a sordid past, highlighted by the misguided Eugenics Movement in the first half of the 20th century, which was another erroneous attempt to apply genetic concepts in the healthcare setting. Unfortunately, eugenicists used pseudo-science in an effort to eliminate certain groups or classes through forced sterilization and other means, in an elitist form of select population control under the guise of improving the genetic pool for society [12]. With that twisted thinking in mind, the Nazis embraced Eugenics and took it to a new level; not only did they attempt to eliminate multiple ethnic groups, but they also used them for gruesome experimentation [12]. These atrocities led to the establishment of the Nuremberg Code. However, those lessons were short-lived when the Tuskegee Syphilis Study, which denied medical treatment or acknowledgment of disease among African American men, was allowed to continue in the name of “science” [12]. Accordingly, it is understandable why a declaration of “race” may be resisted.

With that in mind, the biggest problem with Section 4302(a) of the ACA is the inaccuracy of the terminology. Although the policy’s purpose is essential, to capture healthcare inequality data, it neglects a significant fact. “The completion of the Human Genome Project in 2003 confirmed humans are >99% identical at the DNA level and there is no genetic basis for race” [4] (p. 232). Following the conclusion of the Human Genome Project (HGP), the hope was for a “post-racial society, free of racial prejudice and discrimination” [2]. Its results did not promise to end healthcare disparities; however, its findings should have ended the misconception that “race” was genetic. Unfortunately, the term “race” is still used and leads the general population to believe it has biological validity, especially when used in scientific and medical situations. The general population trusts their clinicians to tell the truth. However, this policy’s wording promotes a lie that divides our country and helps foster discrimination and inequality by reinforcing non-existent genetic differences. Let us begin looking at the evidence with some scientific truths that were only made possible by the trailblazing results of the HGP.

In 2002, Stanford scientists studied human diversity by analyzing 4000 alleles and their correlation to people from different geographical regions. Since all humans have the same genes that code for hair, the varying alleles determine the alternative colors and textures. They determined that “over 92% of alleles were found in two or more regions” and nearly 50% “were present in all seven major geographical regions” [2]. This finding and the lack of uncovering “trademark” alleles corresponding to genetic features of a single group demonstrated the similarity of all humans [2]. Additionally, more discrepancies in alleles exist within a so-called “race” than between them. This concept was highlighted by comparing the DNA of three scientists: two of European descent, James Watson and Craig Venter, and one of Asian descent, Kim Seong-jin. The findings revealed that Watson and Venter only shared one allele, while each shared two with Seong-jin [2]. Using some of the data from the Stanford study, a figure was constructed in a paper with collaboration from Zhivotovsky to demonstrate the human population’s divergence and expansion [13]. This diagram represents one human “race” that migrated to different areas of the globe throughout history, resulting in geographic ancestry, not “race”. 

When reviewing the fine print of the 1997 OMB standards, the government clearly states: “The racial and ethnic categories set forth in the standards should not be interpreted as being primarily biological or genetic in reference. Race and ethnicity may be thought of in terms of social and cultural characteristics as well as ancestry”. However, this fact is not communicated to the populus, and unless they visit this website, it remains unknown to society [9]. Therefore, most Americans think, “If an Asian person looks so different from a European, how could they not be from distinct groups?” [2]. Conceptually, this belief was explained in a recent paper discussing the disturbing concept of genetic essentialism. It is “the belief that people of the same socially defined group (“race” or gender) share genes that make them physically, cognitively, and behaviorally uniform, and distinct from other groups” [3] (p. 1451). In multiple studies, this belief has been shown to result in “racial” discrimination. 

The authors have stated that the prevalence of genetic essentialism is due to the “US Society-like many others-is imbued with implicit and explicit messages reinforcing genetic essentialism and very few messages suggesting otherwise” [3] (pp. 1451–1452). As Allen Goodman shared in the article, Race Is Real, However, It’s Not Genetic, genetic essentialism can be seen in medicine as well, in the story of a woman whose physician refused to send her for bone density testing because “African Americans do not get osteoporosis” [5]. The FRAX calculator used to determine osteoporotic risk requires multiple parameters, including “race”. Additionally, clinicians often discuss risk stratification based on “racial” groups, referring to these differences in health as biological differences. However, genetic variation, not “race”, can explain these differences, as shown previously in the Zhivotovsky study. Human genetic diversity reflects geographic ancestry or local populations, not “race”. Dr. Charles Rotimi highlighted an example of this concept during his presidential address at the 2022 American Society of Human Genetics (ASHG) meeting. He shared that *APOL1* renal risk variants are more prevalent among sub-Saharan Africans due to the gene’s protective advantage against trypanosomiasis. However, due to admixture and gene flow, *APOL1* can be seen in patients who phenotypically appear European or Hispanic. Therefore, if screening is solely based on “race”, many patients may be missed with potentially fatal results [14]. The “racial” differences seen in health are not due to genetics but instead geographic ancestry as well as “systemic differences in lived experience and institutional racism” [5]. Well-balanced diets, decreased access to medical care, and increased societal stress or endemic racism are the root causes of the differences in health outcomes seen within the different ancestral populations [5]. The second example shared by Dr. Rotimi stressed the plight of individuals in Flint, Michigan, and although primarily “Black” neighborhoods, “Whites” who live there are equally affected. Therefore, “living in certain neighborhoods is more important than whether you are Black or White” [14] (p. 399). “Race and genetics versus “race” in genetics: A systematic review on the use of African ancestry in genetic studies”, by Duello et al. is another example of science reinforcing genetic essentialism. It was published in 2021 to highlight how often the scientific community has used “race” as a biological category and why. Unfortunately, over 200 studies have used “race” since 2003 [4]. Finally, the federal government requests “racial” data in censuses and surveys instead of geographic ancestry. So why would the general population not believe “race” is genetic, based on the number of messages they hear from scientific and authoritative sources? 

Even though policymakers believe that using the term “race” to identify different social and cultural characteristics is ideal, criticism about the lack of granularity in the categories of “race” for self-identification persists. The result is gaps in “race” choices, especially for native Americans, native Pacific Islanders, Asians, and descendants of Northern Africa and the Middle East, which creates confusion for patients [7]. The HHS distinctions fail to capture the diversity seen throughout the country [7]. The President’s commission highlighted the recent efforts of the Office of the National Coordinator for Health Information Technology (ONC)’s code set for “race”, including over 900 ways to represent “race”. If implemented, 900 options would introduce even more confusion to an already difficult-to-navigate system [7]. Therefore, the current policy is failing due to confusion over “race”, and the term “race” is erroneous. Yet, health disparities still exist and can only be eradicated through capturing accurate data to guide future corrective action. This leaves policymakers with a serious dilemma.

## 3. New Perspectives

Doing nothing is not an option, since the President’s commission recommends eleven alterations to the existing policy to effectively execute its two-fold goal [7]. Additionally, returning to the policy pre-ACA is not a satisfactory alternative because health disparities exist and they are not improving. The collection of data is imperative to rectify current healthcare inequalities. Data must be collected to analyze trends, make plans, and implement policy changes to improve societal inequalities and reduce health disparities. 

Therefore, option one is to apply the suggested modifications (Table 3), which are lengthy and costly. An example is number 8 in Table 3, which discusses the HHS educating the public on the importance of reporting “race”. This instruction would again propagate the inaccurate belief that there is a biological or genetic distinction between groups of people. The result is the increased dissemination of genetic essentialism, fostering prejudice and “racial” discrimination that would not help us to achieve a post-“racial” society. Additionally, this strategy requires updating the standards again, a method that has not been successful in the past. How would a similar alteration yield a new result? Additionally, adding more “racial” categories, as the commission suggested, greater than 900, to accurately reflect the diverse population will be messy. How can over 900 categories be sorted and organized efficiently?

Option two would replace the word “race” and describe it using genetic ancestry, which is a “valid genomic classifier”, as specified by Dr. Rotimi [14]. He went on to state, “If these centuries of inconsistencies in the meaning of racial categories do not challenge our professional and personal conscience about the use of “race” in research, I am at a loss as to what will do it” [14] (p. 400). Genetic ancestry is defined as evidence regarding the biological descendants of an individual, which is inclusive of their genetic relationships. When used in combination with historical information, the location where distant ancestral relatives inhabited can be determined [15]. Therefore, it is fluid, changing over time, and reflects the migration of populations in the past, present, and future [14]. Unlike “race” or ethnicity, which are static and have failed to accurately capture groups of individuals. As stated clearly by Dr. Rotimi, “descriptors such as Black, White, Hispanic and Asian are at best imprecise proxies to the causal factors that underlay health inequities globally” [14] (p. 399). Scientists are now using the word “Ancestry” to depict human diversity, emphasizing a person’s historical journey instead of attempting to match them to one specific classification. If a scientist wanted to describe the ancestry of a patient with sickle-cell anemia, they might refer to “sub-Saharan African” instead of “black”. Similarly, someone with a diagnosis of cystic fibrosis would be classified as “Northern European” instead of “white” [2]. This option would be revolutionary but is insufficient if we do not improve scientific literacy in the use of genomic information. This transition will require a tremendous amount of education, beginning with children in the K-12 curriculum. Universities and medical schools need to add it to their coursework. Additionally, public service messages would be vital for the community to undo the years of genetic essentialism, as is a re-education of the masses about the truth of the ancestral basis for human diversity. The result is potentially a society with less hesitation to share their origins, making data collection more comprehensive, and health equity attainable.

## 4. Concluding Remarks

Dr. Rotimi spoke passionately about the abolishment of the term “race” and proposed, “I would like to challenge us to ask why we continue to tolerate the use of imprecise labels that we know are hindering our understanding and interpretation of how genetic and non-genetic factors influence human health and identity” [14] (p. 399). By eliminating the word “race” and replacing it with genetically accurate terminology to capture healthcare inequality data, the revolutionary findings of the Human Genome Project, chiefly that “race” is not biological, would translate beyond the world of science to society at large. Although determining the exact terminology falls outside the scope of this paper, the goal here was to unmask the problem. Excluding the word “race” will be a difficult transition, requiring substantial resources to re-educate the population, and commitment from the stakeholders involved to model the change by omitting the word “race” from their vocabulary. Ideally, in a post “race” world, researchers would replace the word “race” with some form of genetic ancestry when capturing demographic information upon entry into clinical trials. Slowly, as the population understands why different groups have distinct appearances and the term is no longer accepted, the Human Genome Project’s discovery will move from the lab to main street and disprove the concept of human “racial” differences.

## Figures and Tables

**Table 1 medicina-59-00861-t001:** “Race” collection through HHS data collection standards adapted from James et al. [7].

Select (X) 1 or More	What Is Your Race?
	White
	2. Black or African American
	3. American Indian or Alaska Native
	4. Asian
	a. Asian Indian
	b. Chinese
	c. Filipino
	d. Japanese
	e. Korean
	f. Vietnamese
	g. Other Asian
	5. Native Hawaiian or other Pacific Islander
	a. Native Hawaiian
	b. Guamanian or Chamorro
	c. Samoan
	d. Other Pacific Islander

**Table 2 medicina-59-00861-t002:** Inconsistencies in “race” collection through health programs adapted from James and colleagues [7].

Setting	Data Collection	Completeness	Self-Reported
Medicare	Varied over time	100%	Yes
Medicaid	HHS 2011	50%	Yes
Marketplaces Fed and State	Fed HHS 2011 State vary	50%	50% of the time
Commercial Insurance	Unknown	<25%	Unknown
Veterans Administration	OMB 1997	100%	50% of the time
Indian Health Service	Blood Quantum and Tribal Affiliation	50%	Unknown
Federally Qualified Health Centers	OMB 1997	100%	Yes
Birth Records	HHS 2011	100%	Yes
COVID-19 Vaccinations	OMB 1997	50%	Unknown
Pregnancy Risk Assessment Monitoring System	OMB 1997	50%	Yes

**Table 3 medicina-59-00861-t003:** Presidential Committee’s recommendations for alterations to Policy 4302(a).

Suggested Modifications to Policy 4302(a)
Update OMB 1997 standards = New population demographics.
2. Require race on Medicare Part C and D applications.
3. Data Working Group needs to standardize data collection.
4. Providers need to reflect communities they serve.
5. Incentives from OMB if federal programs stratify by race.
6. Audits by HHS for gaps in race data for health programs.
7. HHS to provide technical and financial assistance for improved state data systems.
8. HHS to educate the public on the importance of reporting race.
9. Community representatives to endorse race reporting.
10. Congress to encourage data collection and sharing across government systems.
11. Improve privacy protection through health apps.

## Data Availability

Not applicable.

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
