# Peer review of "Race: How the Post-Genomic Era Has Unmasked a Misconception Promoted by Healthcare"

_medicina, 2023, doi:10.3390/medicina59050861_

Round 1

Reviewer 1 Report (New Reviewer)

This is a well-written commentary on an important and a timely topic. The available policies in practice, have been well analysed to show the redundancy of the use of the term "race". However, it would have been more meaningful to the reader if the term "ancestry" and the associated concepts are described in more detail as the concept is a little vague in the write-up. Hence suggest including an introduction to ancestry and how it differs or does not align with race.

Author Response

Thank you so much for your positive feedback. Your insightful and valuable suggestions are greatly appreciated. We will address how we handled each suggestion below and thank you again. We agree there needs to be more of a detailed analysis of the term ancestry especially how it differs from race.We added lines 244-252 in New Perspectives Discussion.

Reviewer 2 Report (New Reviewer)

Dear authors,

I also start my genetics classes by explaining to Portuguese students and those who come from former colonies that there are no human races according to the taxonomic criteria that biologically define race.

I also agree that is important to recognize that language has a powerful impact on people's perception and treatment of others. Using language that is insensitive or offensive can contribute to the marginalization and discrimination of certain groups.

The genomic era came with complicated paradoxes. The genomic information obtained from numerous genotyping projects across the world certainly brought new insights into the genetic differences between individuals and population groups. As a good definition we have: “Precision medicine is based on the idea that by understanding a patient's individual genetic makeup, we can better tailor treatments and interventions to their specific needs. So on an individual level, precision medicine aims to tailor medical treatments to a patient's unique genetic, environmental, and lifestyle factors to improve outcomes and minimize adverse effects. On a population level, precision medicine seeks to identify patterns and biomarkers in large datasets to develop targeted interventions for groups of individuals who share similar genetic and environmental factors, such as certain ethnic or <racial> groups that may be more prone to certain diseases”. On one hand we have to use the term “race” to explain the genomic era outcomes, on the one other hand, the same data show us that in our species we do not differ enough to label the different “population groups” as different “races”, and that the term race would be “obsolete”.

So far, if we ask someone which population group he belongs to, the answer he manages to give is precisely based on previously defined “races” as different population groups.

The truth is that this “taxonomy” does not coincide with a significant genetic variation to define different "races". However, there are many studies that corroborate significant differences between so-called "races", eg. dietetic intolerances, and differences in drug metabolism between individuals of different groups.

I believe that in the future, with the advancement of genomic fields, we will have the possibility to overcome the need to define population groups. There will be so much information available that it will be possible to analyze each individual, with a truly personalized medical approach. In our day genomics technologies, themselves trigger inequalities in human society since they are expensive technologies that are not yet democratized. This is another paradox!

In many areas of society, there is an overstatement of the offensive capacity of certain words and a constant need to define what is politically correct. It seems that now everything has to do with the sensitivities of a certain group of the population. Removing the word "race" from this area where we all know what it means, will not stop racism or inequalities that some groups of the population are subjected to.

I think that in our days we still need this categorization in order to avoid missing important data.

I really like your article. I think that you must improve the discussion with other examples and points of view and data about differences between "races".

Kind regards.

Author Response

Thank you so much for your encouraging feedback. Your commentary was valuable and thoughtful. We will address how we handled each suggestion below and thank you again.

We agree there needs to be more examples sighting the differences between 'races'. We added examples from literature as well as cited from the Presidential address at the ASHG annual meeting 2022- Lines 154-158, 186-192, and 197-200.

Reviewer 3 Report (New Reviewer)

Comment to Authors

Title: Race: How the Post-Genomic Era Has Unmasked a Misconception Promoted by Healthcare

Main Comment

The term “race” has been a controversial term social although biological information and advancement in technology has unraveled some these misconceptions. Schaare et al, in this paper sought to review the term ‘Race” using current information on human genome and also review the current policy on the term race, its limitations with recommendation. The author

The authors have put in a good effort to dispel the notion of the use of race in biological and genetic context. This is agood paper and will strongly encourage the author to take time and read over the manuscript again to make all necessary corrections.

The conclusions derived from the study are valid since they were use existing history and information on race and health care policies. The paper is recommended for publication subject to minor corrections. The authors need to state that this review and paper was prepared in the score of the USA Affordable Care Act and health care system as such may not reflect other regions such are Africa, Asia and Middle East Health Care systems.

The arguments presented in this review are very important for policy regarding race and heath access as well as other societal implication. I highly recommend the paper to be published but on condition that the author state in the abstract that the write up is heavy leaned on the US system on health care and policies concerning biological diversity of individuals in society. Also the authors should clearly define and differentiate the terms “race” and “ethnicity” comprehensively to make their case strong.

Minor Comments

Abstract

Introduction

Line 34 Kindly change biologic to physiological genetic factors are also biological factors

Discussion

The discussion presented are sound and compelling to support the notion of the authors.

Conclusion

The conclusions derived are valid.

Author Response

Thank you so much for your positive feedback. Your insightful recommendations for revisions were greatly appreciated. We will address how we handled each suggestion below and thank you again.

We agree there needs to be a disclaimer that the paper was written in the context of the US Healthcare system and the Affordable Care Act. We added lines 16-18 in the abstract and lines 96-99 in the first paragraph of the policy analysis section. Also, we agreed there was a need for a better definition of race and ethnicity compared to genetic ancestry to improve the overall argument-Lines 244-252.

Line 34 was changed to physiologic genetic factors to reflect your suggestion.

This manuscript is a resubmission of an earlier submission. The following is a list of the peer review reports and author responses from that submission.

Round 1

Reviewer 1 Report

In their manuscript, Schaare et al. review the history of the term “race” and discuss current policies focusing on the correlation between health care services and geographic ancestry. Further, they propose alterations to improve the current situation.

This commentary is a valuable contribution to the scientific community as it challenges common terminology and encourages replacing the concept of “race” with alternatives reflecting current scientific knowledge. Thus, I have only a few minor comments.

The paragraphs introduction and Policy analysis are, in parts, hard to read and understand for a non-US audience. I thus encourage the authors to re-write these paragraphs to provide a rather general view or, for example, explain the scope of duties of some of the mentioned institutions (e.g. OMH, HHS, OMB).

The authors suggest using the concept of geographic ancestry instead of “race” since biological and genetic differences might be essential in medical treatment. However, its repeated use within the manuscript replicates the term. As such, I suggest using “race” in the manuscript consistently in quotation marks.

Author Response

Reviewer 1:

Thank you so much for your positive feedback. Your insightful and valuable suggestions are greatly appreciated. We will address how we handled each suggestion below and thank you again!

“The paragraphs’ introduction and Policy analysis are, in parts, hard to read and understand for a non-US audience. I thus encourage the authors to re-write these paragraphs to provide a rather general view or, for example, explain the scope of duties of some of the mentioned institutions (e.g. OMH, HHS, OMB).”

This was a very important point that we had overlooked. We needed to address an international audience. Therefore, we went back to the introduction and explained the responsibilities of each institution. There are now definitions of the HHS, OMH, OMB, and the Agency for Healthcare Research and Quality.

“The authors suggest using the concept of geographic ancestry instead of “race” since biological and genetic differences might be essential in medical treatment. However, its repeated use within the manuscript replicates the term. As such, I suggest using “race” in the manuscript consistently in quotation marks.”

We agree on the potential confusion. We went back and changed any mention of ‘race’ to include quotation marks and even added quotation marks to the term ‘racial’.

Reviewer 2 Report

The authors indicate they will review the history of race, analyze current policy on the use of race, discuss its limitations, and make recommendations for alterations.  It is not clear how genomics is linked to the ‘lie’.  Are the authors predicating the Human Genome Project promised to address health disparities, though most causes of health disparities are socioeconomic and sociopolitical?  Are the authors stating that all collection of racial//ethnic data erroneously furthers the concept of race as a biological category?  If the Affordability Care Act had not collected racial data, how could the government assess whether it addressed the huge burden of lack of insurance to the racial/ethnic populations who bear a huge burden of poor health?

• The history of race has been discussed in multiple publications, most notably ‘The Race Myth’ and ‘The Emperor’s New Clothes’ by Joseph Graves, Jr. 

•. A statement of the  Presidential Committee’s recommendations for alterations of Policy 4302(a) is summarized in Table 3, but not discussed.

•. The limitations do not explain a ‘big lie’.

•. The substitution of ancestry for race is inadequate.  The use of ‘genetic ancestry’ is distinct from ‘genealogy’, but still does not encompass that inheritance is only from direct ancestors.

The ‘big lie’ in healthcare.

1.      It is not clear what the ‘big lie’ is/was as the findings of the Human Genome Project as cited in the academic literature did not promise an end to health inequity and health disparities.  The vast majority of health disparities and healthcare inequities are due to socioeconomic and sociopolitical reasons.  Not biological or genetic reasons.  The findings of the Human Genome Project raised hope for genetic solutions to disease.  It did not/could not promise an end to socioeconomic or sociopolitical causes of health disparities, nor the promise of health equity. 

2.     Every use of race does not imply it is a biological category.  Authors need to clarify why the collection of racial data is important and that its use needs to be explained/justified.

3.     The authors cite National Geographic for a definition of race, rather the peer-reviewed scientific literature. 

Author Response

Reviewer 2:

Thank you very much for the insightful and valuable feedback. It was greatly appreciated. We will address how we handled each suggestion below and thank you again!

“The authors indicate they will review the history of race, analyze current policy on the use of race, discuss its limitations, and make recommendations for alterations.  It is not clear how genomics is linked to the ‘lie’.  Are the authors predicating the Human Genome Project promised to address health disparities, though most causes of health disparities are socioeconomic and sociopolitical?  Are the authors stating that all collection of racial//ethnic data erroneously furthers the concept of race as a biological category?  If the Affordability Care Act had not collected racial data, how could the government assess whether it addressed the huge burden of lack of insurance to the racial/ethnic populations who bear a huge burden of poor health?” We realized that we needed to further clarify that the collection of ‘racial’ data erroneously furthers the concept of ‘race’ as a biological category. We added 2 sentences to the end of the first paragraph of the introduction which should better explain why the word is scientifically inaccurate, hence the ‘lie’. We also, clarified that we were not implying that the HGP promised to end healthcare disparities (added a sentence to the middle of the 2nd paragraph after Table 2 on page 4) and that healthcare disparities are a result of socio-economic and socio-political causes (added wording to the 1st sentence of the 2nd paragraph of the Introduction). Finally, we reinforced the importance of collecting healthcare inequity data in order to improve outcomes for all (added wording to the 2nd sentence of the 2nd paragraph after Table 2 on page 4, added a sentence to the middle of the 1st paragraph in New Perspectives, and added wording in the first sentence of the conclusion).

  • “The history of race has been discussed in multiple publications, most notably ‘The Race Myth’ and ‘The Emperor’s New Clothes’ by Joseph Graves, Jr.” We agree with the reviewer’s comments, but we believe the cultural aspects of the concept of ‘race’ have been embedded in the tapestry of the US society since its formation as a nation. We just want to focus on the genetic aspects of the concept of ‘race’ and therefore, we believe the discussion of the cultural aspects of ‘race’ is beyond the scope of this paper.

  • . “A statement of the  Presidential Committee’s recommendations for alterations of Policy 4302(a) is summarized in Table 3, but not discussed.” We agree on the need for an example to be discussed. We added an example of the 8th recommendation from the table in the 2nd paragraph in New Perspectives.

  • . “The limitations do not explain a ‘big lie’.” As stated earlier, we agreed that there may have been a lack of clarity so we added 2 sentences to the end of the first paragraph of the introduction which should better explain why the word is scientifically inaccurate, hence the ‘lie’.

  • . “The substitution of ancestry for race is inadequate.  The use of ‘genetic ancestry’ is distinct from ‘genealogy’, but still does not encompass that inheritance is only from direct ancestors.” We agree that ancestry is inadequate to replace ‘race’. We made multiple changes to correct this implication. (Added wording to the first paragraph after Table 3, a sentence to the middle of the same paragraph, and wording to the 3rd sentence of the conclusion)

“The ‘big lie’ in healthcare.”

  1. “It is not clear what the ‘big lie’ is/was as the findings of the Human Genome Project as cited in the academic literature did not promise an end to health inequity and health disparities.  The vast majority of health disparities and healthcare inequities are due to socioeconomic and sociopolitical reasons.  Not biological or genetic reasons.  The findings of the Human Genome Project raised hope for genetic solutions to disease.  It did not/could not promise an end to socioeconomic or sociopolitical causes of health disparities, nor the promise of health equity.” We agree on the potential confusion and made the following changes:

We realized that we needed to further clarify that the collection of ‘racial’ data erroneously furthers the concept of ‘race’ as a biological category. We added 2 sentences to the end of the first paragraph of the introduction which should better explain why the word is scientifically inaccurate, hence the ‘lie’. We also, clarified that we were not implying that the HGP promised to end healthcare disparities (added a sentence to the middle of the 2nd paragraph after Table 2 on page 4) and that healthcare disparities are a result of socio-economic and socio-political causes (added wording to the 1st sentence of the 2nd paragraph of the Introduction). Finally, we reinforced the importance of collecting healthcare inequity data in order to improve outcomes for all (added wording to the 2nd sentence of the 2nd paragraph after Table 2 on page 4, added a sentence to the middle of the 1st paragraph in New Perspectives, and added wording in the first sentence of the conclusion).

  1. “Every use of race does not imply it is a biological category.  Authors need to clarify why the collection of racial data is important and that its use needs to be explained/justified.” We agree with your comments and added: we reinforced the importance of collecting healthcare inequity data in order to improve outcomes for all (added wording to the 2nd sentence of the 2nd paragraph after Table 2 on page 4, added a sentence to the middle of the 1st paragraph in New Perspectives, and added wording in the first sentence of the conclusion).
  2. “The authors cite National Geographic for a definition of race, rather the peer-reviewed scientific literature.” We agree and appreciate the suggestion. The definition was changed in the first sentence of the first paragraph of the Introduction using a peer-reviewed publication and omitting the National Geographic reference.

Round 2

Reviewer 2 Report

Substantive changes have not been made.

Other corrections:

Line 30:   . . .  has been defined by Ford and Kelly as “a social construct  . . .

Line 44:  Health disparities include biological and genetic determinants.  Not only socioeconomic or sociopolitical.

Line 58:  Omit sentence > The OMB does not monitor the effectiveness of all government programs.

Line 73:  Designed, not ‘intending’.

Line 144:  Misconception, not concept.

Line 221:  Not if the rationale for collecting racial information was explained.

Line 224; j Recommend a term.

Author Response

Dear Reviewer,

Thank you very much for your continued feedback, it is appreciated.  Below describes the changes we made based on your suggestions.

Are the results clearly presented- Must be improved. Are the conclusions supported by the results- Must be improved. Substantive changes have not been made. Since this is a commentary type of paper, it does not contain any results or conclusions. The Commentary articles allow authors to contribute viewpoints on the interpretation of recent findings, the value, and the weaknesses and strengths of hypotheses. So, it is only a review of the recent findings, not new results of our own research.

Other corrections:

Line 30:   . . .  has been defined by Ford and Kelly as “a social construct  . . .  The title of the article was omitted. Thank you.

Line 44:  Health disparities include biological and genetic determinants.  Not only socioeconomic or sociopolitical.  We agree that historically health disparities have been linked to biological and genetic factors, which we added to the first sentence of the second paragraph. Thank you.

Line 58:  Omit sentence > The OMB does not monitor the effectiveness of all government programs. The first reviewer requested that we add that sentence to the paper, so we are unable to remove it.

Line 73:  Designed, not ‘intending’. Thank you for the recommendation. We made the change.

Line 144:  Misconception, not concept. Again, thank you for the correction. We altered the text accordingly.

Line 221:  Not if the rationale for collecting racial information was explained. We only implied genetic rationale as invalid.

Line 224; j Recommend a term. We suggested origin and added it to the sentence. Thank you again for all your feedback.
